# A capture enzyme-linked immunosorbent assay for detection of mosquito salivary protein-specific immunoglobulin E

Zhaoyang Wang[1,2,3☯], Yan Liang[1☯], Fen Zeng[4], Tingting Li[4*], Gong Cheng[ID][1,2,3,5*]

**1** New Cornerstone Science Laboratory, Tsinghua University-Peking University Joint Center for Life Sciences, School of Basic Medical Sciences, Tsinghua University, Beijing Key Laboratory of Viral Infectious Diseases, Beijing, China, **2** Institute of Infectious Diseases, Shenzhen Bay Laboratory, Shenzhen, China, **3** Institute of Pathogenic Organisms, Shenzhen Center for Disease Control and Prevention, Shenzhen, China, **4** People's Hospital of Xishuangbanna Dai Autonomous Prefecture, Jinghong, Yunnan, China, **5** Southwest United Graduate School, Kunming, China

☯ These authors contributed equally to this work.
* gongcheng@mail.tsinghua.edu.cn (GC); xsbnltt@163.com (TL)

## Abstract

Mosquito saliva contains numerous distinct mosquito salivary proteins (MSPs) that mediate mosquito-host interactions. Repeated mosquito exposure can trigger allergic reactions, with MSP-specific IgE playing a central role. Current enzyme-linked immunosorbent assay (ELISA) and immunoblotting methods for detecting MSP-specific IgE suffer from interference by much more abundant MSP-specific IgG, leading to low sensitivity. Here, we developed a capture ELISA to overcome these limitations. We compared the performance of this capture ELISA with the conventional indirect ELISA in detecting MSP-specific IgE titers in sera from both repeatedly exposed mice and human volunteers. The results demonstrated that, compared to the indirect ELISA, the novel capture ELISA exhibited significantly superior sensitivity and specificity. Using serum samples from 20 volunteers with confirmed exposure to *Aedes aegypti* bites and 20 volunteers without such exposure, we found that the capture ELISA achieved 100% diagnostic sensitivity and specificity (20/20), with both false-positive and false-negative rates at 0% (0/20). The limit of detection was determined to be 87.42 ng/mL total IgE equivalent in human serum. Furthermore, we dynamically monitored *Aedes aegypti* salivary protein AAEL000749-specific IgE titers in healthy individuals from areas with widespread mosquito distribution using the capture ELISA. The results showed that both the positive rate and titer of AAEL000749-specific IgE in the sera were significantly higher during months with elevated mosquito population densities, compared to months with lower densities. This indicates that, under natural exposure conditions, the levels of MSP-specific IgE in human sera are closely correlated with local mosquito densities. In summary, our novel capture ELISA demonstrates excellent diagnostic performance and can be used for the quantitative analysis of MSP-specific IgE in mammalian sera. This provides a powerful tool

**Data availability statement:** All relevant data are stored in the Zenodo database and are accessible at: https://doi.org/10.5281/zenodo.15321036.

**Funding:** This study was supported by grants from the National Key Research and Development Plan of China (2023YFA1801000) to GC, Shenzhen Medical Research Fund (B2404002, B2402011) to GC, the National Natural Science Foundation of China (32188101, 82422049) to GC, the Shenzhen San Ming Project for Prevention and Research on Vector-borne Diseases (SZSM202211023) to GC, the Science and Technology Project of Southwest United Graduate School of Yunnan (202302A0370010) to GC, the Science and Technology Program of Yunnan Provincial Department of Science and Technology (202105AC160007) to TTL, the New Cornerstone Science Foundation through the New Cornerstone Investigator Program to GC, and the XPLORER PRIZE to GC. The funders had no role in study design, data collection and analysis, decision to publish, or preparation of the manuscript.

**Competing interests:** The authors have declared that no competing interests exist.

for the diagnosis and prognosis of mosquito allergy, as well as for monitoring mosquito exposure levels in endemic areas.

## Author summary

When mosquitoes bite host animals, they secrete hundreds of salivary proteins. These proteins serve as a bridge for interactions between mosquitoes and their hosts. Numerous studies have shown that mosquito salivary proteins facilitate viral infection at the bite site. In addition, mosquito salivary proteins are highly immunogenic and can stimulate both non-specific and specific immune responses in the host. Symptoms such as wheals and itching caused by mosquito bites are manifestations of skin allergic reactions induced by these salivary proteins. The specific IgE antibodies against mosquito salivary proteins produced in the host are key mediators of mosquito allergy and are important for the pathogenesis, diagnosis, and prognosis of mosquito allergic reactions. However, detection of mosquito salivary protein-specific IgE is challenging, mainly due to interference from mosquito salivary protein-specific IgG present in host serum. Here, we report a capture ELISA-based method for detecting mosquito salivary protein-specific IgE. Compared with traditional methods, this assay offers higher sensitivity and specificity, allows for quantitative analysis, and thus provides an efficient approach for detecting mosquito salivary protein-specific IgE in mouse or human serum.

## Introduction

Repeated mosquito exposure induces mosquito salivary protein (MSP)-specific immunoglobulin E (IgE), which can sensitize host cutaneous mast cells [1]. When the mosquito bites the host again, the secreted MSPs bind to the specific IgE on the surface of sensitized host cutaneous mast cells, leading to their activation. Activated cutaneous mast cells release a variety of proinflammatory mediators, causing immediate hypersensitivity, which is manifested as erythema or wheals at the mosquito bite site, causing itching or pain and other symptoms [2]. Almost all people have an allergic reaction to MSPs after repeated mosquito bites, but most people's allergic reactions are transient and disappear within a few hours [3]. Some people have a significantly enhanced or prolonged allergic reaction to mosquitoes, called mosquito allergy [4]. Mosquito allergy seriously affects the quality of life of patients, and even threaten their lives [4]. At present, it is recognized that MSP-specific IgE plays a key role in the pathogenesis of mosquito allergic reaction and mosquito allergy. Thus, the detection of MSP-specific IgE in host serum is of great significance for the diagnosis and prognosis of mosquito allergy.

Although MSP-specific IgE plays an important role in the pathogenesis, diagnosis and prognosis of mosquito allergy, current methods for the detection of MSP-specific

IgE are inefficient. There are two main causes for this poor test efficiency: 1) It is quite difficult to collect enough MSPs, and even more difficult to prepare an individual MSP. 2) The concentration of IgE is much lower than that of IgG in host serum, making it difficult to exclude the interference of MSP-specific IgG.

There are several methods for the detection of MSP-specific IgE, mainly including enzyme-linked immunosorbent assay (ELISA) [5] and immunoblotting tests [6–8]. Three limitations lead to very limited use of immunoblotting test. 1) the interference of MSP-specific IgG. 2) the requirement of large amount of MSPs. 3) the inability to accurately quantify the titer of MSP-specific IgE.

ELISA is a powerful tool to detect specific antigens or antibodies with high detection sensitivity and specificity [9]. At present, most of the ELISA methods for detecting MSP-specific IgE are indirect ELISA [10–12]. Although these methods are easy to operate, they cannot exclude the interference of serum IgG, either. When MSP-specific IgG and IgE coexist, immobilized MSPs will preferentially be occupied by the specific IgG due to its superior concentration [13,14]. As a result, the detection sensitivity of indirect ELISA is significantly lower than capture ELISA. Capture ELISA can specifically enrich IgE while remove IgG, thus significantly increasing the sensitivity and specificity of detection. However, the capture ELISA also requires the use of a large amount of MSPs [15], which makes the whole operation quite time-consuming and thus limits its application. The laborious collection of MSPs from mosquitoes can be solved by using *in vitro* expression system, but the expression system described in the current literature is prokaryotic [16], the MSPs produced in prokaryotic system may have altered structure and function from the mosquito natively-synthesized MSPs, which may affect the detection of MSP-specific IgE. In this study, we used the *Drosophila* S2 cell expression system to express two *Aedes aegypti* salivary proteins AAEL0006347 and AAEL0000749 and purified them by using 6x His tag. MSPs expressed in *Drosophila* S2 cells can ensure the same structure and function as the mosquito natively-synthesized salivary proteins.

Our capture ELISA is a deep optimization and improvement of the currently available capture ELISA. It has the following four advantages. 1) A cheaper and more specific monoclonal antibody against the Fc fragment of human and mouse IgE is used as the capture antibody, which not only increases the capture efficiency of IgE, but also well keeps the binding capacity of IgE F(ab)$_2$ fragments to MSPs. 2) The highly pure MSPs expressed by *Drosophila* S2 system are used to replace the MSPs obtained by collecting mosquito saliva, thus reducing the time-consuming and laborious collection step of mosquito saliva. 3) A V5 tag was added to the carboxyl terminus of each recombinant MSP so that a monoclonal antibody against the V5 tag can be used to detect all the individual MSPs, and use of a monoclonal antibody can improve the specificity of detection. 4) An HRP-labeled anti-V5 tag IgG antibody is used to amplify the detection signal and an ultra-sensitive TMB chromogenic substrate is used to further enhance the detection sensitivity.

## Methods

### Ethics statement

The human blood used for detection of mosquito salivary protein (MSP)-specific IgE in serum was donated by healthy volunteers and collected at Tsinghua University Hospital and Xishuangbanna Dai Autonomous Prefecture People's Hospital. All the volunteers had given formal written consents. The Medical Ethics Committee of Tsinghua University (approval number: THU01-20240062) and the Medical Ethics Committee of Xishuangbanna Dai Autonomous Prefecture People's Hospital (approval number: 2025004) approved the collection and use of human blood samples.

### Mice and mosquitoes

C57BL/6J (Cat# Jax 000664, RRID:IMSR_JAX:000664, The Jackson Laboratory) mice were purchased from The Jackson Laboratory and bred at Tsinghua University. The mice were bred and maintained under a specific pathogen-free animal facility at Tsinghua University. Four-week-old female C57BL/6J mice were used for the animal studies. All experiments were approved by and performed under the guidelines of the Experimental Animal Welfare and Ethics Committee of Tsinghua

University (approval number: 25-CG1). *Aedes aegypti* mosquitoes (Rockefeller strain) were reared at 28 °C and 80% humidity in a specially designed incubator (Cat# Model 818, Thermo Fisher) following standard rearing procedures [17].

## Repeated mosquito bites on mice

To induce MSP-specific antibodies, C57BL/6J mice were bitten by uninfected *Aedes aegypti* mosquitoes twice weekly for 4 consecutive weeks. During each session of mosquito bites, C57BL/6J mice were sedated with approximately 250 mg/kg of tribromoethanol in a single intraperitoneal dose. Sedated mice were placed on mosquito cups to allow for mosquito biting from 1, 5, or 10 mosquitoes per mouse, and the unbitten mice served as negative controls. Feeding was confirmed by visualization of blood within abdomen of the mosquitoes. Mouse sera were collected at 7, 14, 21, and 28 days post mosquito exposure.

## Immunization of mice with MSPs

Four-week-old female C57BL/6J mice were randomly assigned to four groups, with eight mice per group: the AAEL000749-immunized group, the AAEL006347-immunized group, the bovine serum albumin (BSA, Cat# V900933, Sigma)-immunized group, and the adjuvant-only control group. Five micrograms of MSPs or BSA were mixed with an equal volume of aluminum adjuvant (Cat# vac-alu-250, InvivoGen) and administered subcutaneously to the mice in the respective groups. A booster immunization was given two weeks later. Serum samples were collected seven days after the second immunization.

## Repeated mosquito bites on healthy volunteers

Four healthy adult volunteers including one with- and three without- prior history of *Aedes aegypti* mosquito exposure were recruited. Each volunteer underwent controlled bites by uninfected *Aedes aegypti* mosquitoes twice weekly for 4 consecutive weeks. During each session, 10 uninfected *Aedes aegypti* mosquitoes starved for 24 h were confined in a mesh-covered feeding apparatus ($20 \times 20 \times 20$ cm$^3$) with the volunteer's forearm exposed to mosquitoes for 5 min to allow feeding. Feeding was confirmed through visualization of blood within the abdomen of the mosquitoes. Bite induced adverse effects were monitored throughout the study, and the volunteers could withdraw from the study at any time they wanted.

## Recombinant MSP purification

The cDNA derived from the mosquito salivary gland was used to amplify the *Aedes aegypti* salivary protein-coding genes, which were then inserted into the pMT/Bip/V5-HisA vector (Cat# V4130-20, Invitrogen) to construct the salivary protein plasmids. *Drosophila melanogaster* S2 cells were co-transfected with the salivary protein plasmids and a hygromycin-resistant plasmid at 19:1 ratio. A Schneider's Drosophila Medium (Cat#21720-024, Gibco) supplemented with 10% fetal bovine serum (Cat# 16000-044, Gibco) and 300 µg/mL hygromycin (Cat# H8080, Solarbio) was used to select and maintain the stably transfected cell lines. Expression of the recombinant MSPs was induced by culturing cells in Express Five Serum Free Medium (10486-025, Gibco) supplemented with 500 µM copper sulfate (HY-Y1878C, MCE). After four days of induction, the culture supernatant, which contained abundant recombinant MSPs, was collected and incubated with cobalt beads (Cat# 635515, Clontech) at 4 °C overnight. The recombinant MSPs were C-terminally labeled with a V5 tag and a 6x His tag and thus could be purified by using cobalt beads. After incubation with the culture supernatant, the cobalt beads were loaded onto a column and sequentially washed with 100 mL phosphate-buffered saline (PBS), 50 mL 2.5 mM imidazole (Cat# I811845, Macklin) in PBS, and 20 mL 5 mM imidazole in PBS. The bound proteins were then eluted stepwise with 15 mL 10 mM imidazole in PBS, 15 mL 20 mM imidazole in PBS, and 10 mL 250 mM imidazole in PBS. The recombinant MSPs present in each eluted fraction were analyzed by sodium dodecyl sulfate-polyacrylamide gel

electrophoresis (SDS-PAGE) followed by Coomassie brilliant blue (Cat# G4540, Solarbio) staining. Fractions containing only the recombinant MSPs were pooled, concentrated, and buffer-exchanged into PBS using ultrafiltration tubes (Cat# UFC901096, Merck). The concentration of purified recombinant MSPs was determined with a Bradford Protein Assay Kit (Cat# P0006, Beyotime) according to the manufacturer's instructions. The final recombinant MSP preparations were aliquoted and stored at -80 °C. The identity of the purified recombinant MSPs were validated by immunoblotting analysis of the V5 tag and checked for purity with SDS-PAGE and Coomassie brilliant blue staining [18].

## Detection of MSP-specific IgE by indirect ELISA

To detect MSP-specific IgE by indirect ELISA, high-binding microplates (Cat# 3690, Corning) were loaded at 8 µg/mL (50 µL/well) with purified MSP in 1x ELISA coating buffer (composed of 1.59 g $Na_2CO_3$, 2.93 g $NaHCO_3$, and deionized water to a total volume of 1 L) over night at 4°C, followed by blocking using 5% (wt/vol) milk (Cat# 232100, BD) in PBS for 2 h at 25°C. Serum was diluted in 0.5% milk at appropriate dilutions (1:5 for human serum and 1:10 for mouse serum) and the plates were incubated with 50 µL of the diluted serum for 2 h at 25°C. After the plates were washed with PBS containing 0.02% (vol/vol) Tween 20 (Cat# P1379, Sigma) (PBST), 50 µL of a 1/1,000 dilution of HRP-labeled goat anti-mouse IgE secondary antibody (Cat# SA5-10263, RRID:AB_2868311, Invitrogen) was added to each well and the plates were incubated for 1 h at 25°C. After being washed with PBST, the bound antibodies were detected by incubation with 50 µL ultrasensitive TMB chromogen solution (Cat# P0206, Beyotime). After incubation for 10 min at 25°C, 50 µL ELISA Stop Solution (Cat# C1058, Solarbio) was added and OD value was determined at a wavelength of 450 nm (OD450). The ΔOD450, defined as the OD450 value of the test serum minus the mean OD450 value of the negative control sera, was used to indicate the titer of AAEL000749-specific IgE in serum samples. For mouse sera, the negative control sera were obtained from mice of the same age, gender, and housing conditions that had not been exposed to mosquito bites. For human sera, the negative control sera were collected from healthy volunteers who had never been bitten by *Aedes aegypti* (for details, see "Detection of MSP-specific IgE by capture ELISA").

## Detection of MSP-specific IgE by capture ELISA

To detect MSP-specific IgE by capture ELISA, high-binding microplates were loaded at 4 µg/mL (50 µL/well) with an IgE-capture monoclonal antibody Omalizumab (Cat# HY-P9950, RRID:AB_3695275, MCE) for human serum or rabbit anti-mouse IgE antibody (Cat# RAM/IgE(Fc)/7S, Nordic-MUbio) for mouse serum in 1x ELISA coating buffer (composed of 1.59 g $Na_2CO_3$, 2.93 g $NaHCO_3$, and deionized water to a total volume of 1 L) over night at 4°C, followed by blocking with 5% milk in PBS for 2 h at 25°C. Serum was diluted in 0.5% milk at appropriate dilutions (1:5 for human serum and 1:10 for mouse serum) and the plates were incubated with 50 µL of the diluted serum for 2 h at 25°C. After the plates were washed with PBST, 50 µL of 0.1 µg/mL of purified MSP AAEL000749 (labeled with a V5 tag) in 0.5% milk was added to each well and the plates were incubated for 1 h at 25°C. After the plates were washed with PBST, 50 µL of a 1/2,000 dilution of anti-V5-tag antibody (Cat# M167-3, RRID:AB_1953024, MBL) in 0.5% milk was added to each well and the plates were incubated for 1 h at 25°C. After being washed with PBST, 50 µL of a 1/5,000 dilution of HRP-labeled goat anti-mouse IgG antibody (Cat# 330, RRID:AB_2650507, MBL) in 0.5% milk was added to each well and the plates were incubated for 40 min at 25°C. After being washed with PBST, 50 µL ultrasensitive TMB chromogen solution was added to each well. After incubation for 10 min at 25°C, 50 µL ELISA Stop Solution was added and OD450 value was determined at a wavelength of 450 nm.

The ΔOD450, defined as the OD450 value of the test serum minus the mean OD450 value of the negative control sera, was used to indicate the titer of AAEL000749-specific IgE in serum samples. The negative control sera were obtained from eight healthy volunteers (4 males and 4 females, aged 18–30 years) recruited at Tsinghua University. All volunteers had lived long-term in northern China, a region where *Aedes aegypti* is not distributed. Therefore, these individuals are theoretically unlikely to have

been bitten by *Aedes aegypti*, and their sera are expected to lack IgE antibodies specific to *Aedes aegypti* MSPs. Thus, these samples serve as appropriate negative control sera. The cut-off value for determining the positivity of AAEL000749-specific IgE antibodies was set at twice the maximum ΔOD450 ($NC_{max}$) value of the negative control sera (i.e., 2-fold of $NC_{max}$).

The grading criteria for MSP-specific IgE titers were as follows: negative, $\Delta OD450_{sample} < 2$-fold of $NC_{max}$; low titer, $\Delta OD450_{sample} \geq 2$-fold of $NC_{max}$ but $< 5$-fold of $NC_{max}$; medium titer, $\Delta OD450 \geq 5$-fold of $NC_{max}$ but $< 10$-fold of $NC_{max}$; and high titer, $\Delta OD450 \geq 10$-fold of $NC_{max}$.

### Determination of total IgE concentration in human serum samples

An ELISA kit (Cat# E-EL-H6104, Elabscience) was used for the quantification of human serum IgE according to the manufacturer's instruction. Briefly, serial two-fold dilutions of the human IgE standard were prepared and measured by ELISA in parallel with the human serum samples to be tested (diluted 1:500). A standard curve was constructed by plotting the OD450 values of the standards against their concentrations, and a linear regression equation was generated. The total IgE concentration in the test human serum samples was then calculated from their OD450 values using the regression equation.

### Statistical analysis

Quantitative data that conformed to normal distribution and had equal variance were presented as the mean ± SEM. GraphPad Prism 10.0.0 software was used for statistical analyses. One-way analysis of variance (ANOVA), two-way ANOVA and post-hoc multiple *t* tests were used to compare data with three or more groups. Statistical significance was set at $p < 0.05$. *n.s.* denotes not significant, * denotes $p < 0.05$, ** denotes $p < 0.01$, and **** denotes $p < 0.0001$.

## Results

### Rationales and operating procedures of capture ELISA and indirect ELISA

Fig 1A shows the rationale and operating procedures of indirect ELISA. First, MSPs were coated on the bottom of a high affinity microplate at a concentration of 8 µg/mL overnight at 4 °C, followed by blocking with 5% skim milk for 2 hours at room temperature. For detection, serum was diluted with an appropriate ratio (1:5 or 1:10) and incubated for 2 hours at room temperature, followed by incubation with an HRP-labeled anti-IgE antibody for 1 hour at room temperature. Finally, TMB substrate was added for colorimetry (Fig 1A). Notably, MSP-specific IgG in serum is much higher in concentration than MSP-specific IgE, allowing the MSP-specific IgG to preferentially bind the coated MSPs, thereby blocking MSP-specific IgE binding and significantly reducing the sensitivity of indirect ELISA (Fig 1A). Fig 1B shows the rationale and operation procedures of our capture ELISA. First, a monoclonal capture antibody against IgE was coated on the bottom of a high affinity microplate at a concentration of 4 µg/mL overnight at 4°C, followed by blocking with 5% skim milk for 2 hours at room temperature. For detection, serum was diluted with an appropriate ratio (1:5 or 1:10) and incubated for 2 hours at room temperature. After that, an individual MSP with a V5 tag was added at 0.1 µg/mL and incubated for 1 hour at room temperature. And then a mouse anti-V5 tag IgG was added and incubated at room temperature for 1 hour. After that, HRP-labeled anti-mouse IgG was added and incubated at room temperature for 40 min. Finally, a TMB substrate solution was added for colorimetry.

Although the operating procedures of the capture ELISA is slightly more complicated, by specifically capturing IgE in the serum to be tested, it can not only enrich IgE, but also eliminate the interference of IgG on the detection of IgE, and finally drastically increase the sensitivity and specificity of the capture ELISA.

### Comparison of the effectiveness of capture ELISA and indirect ELISA in detecting MSP-specific IgE in mouse serum

First, we established the mouse model of repeated mosquito bites, which could induce the production of specific IgG and IgE antibodies against MSPs (Fig 2A). The protein encoded by the *AAEL000749* gene (designated AAEL000749) is

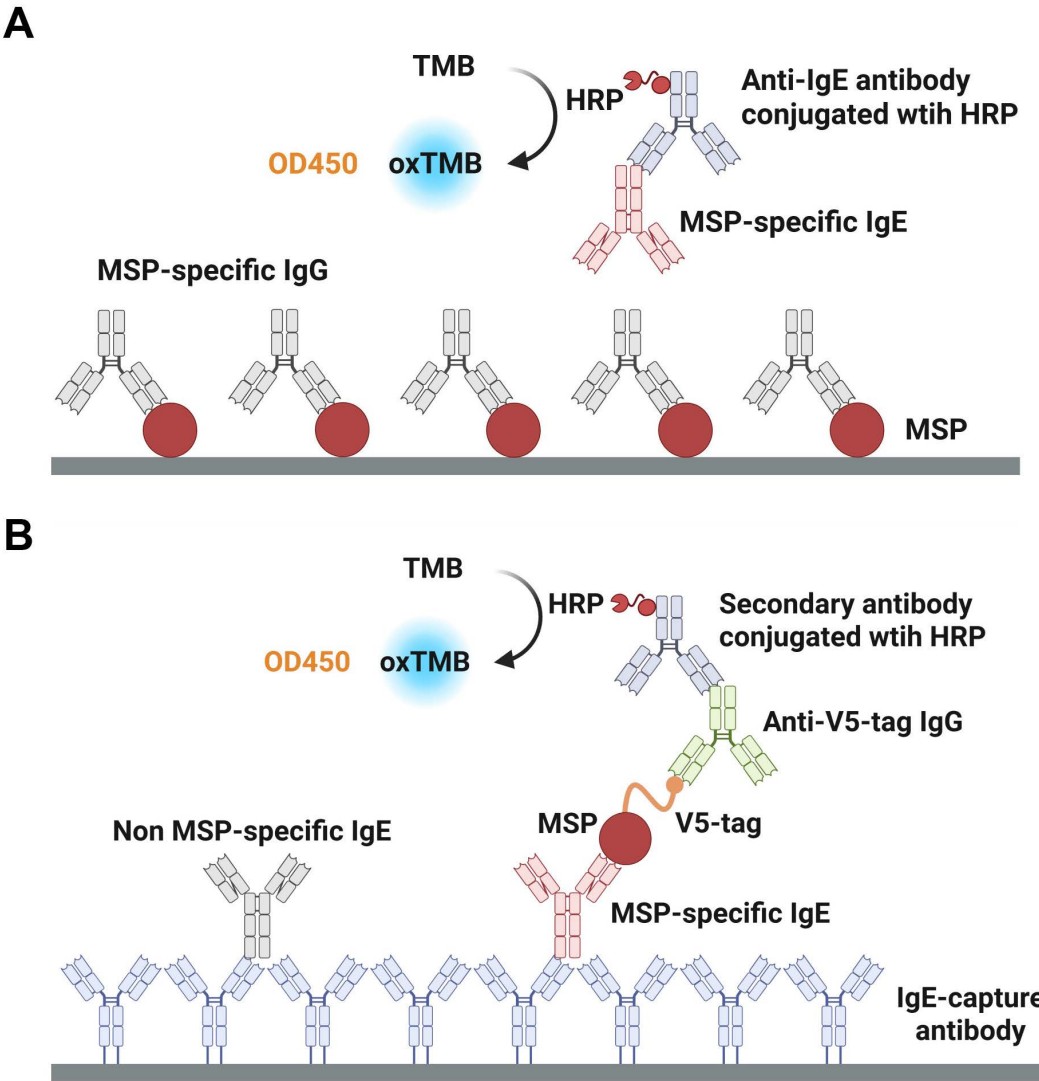

**Fig 1. The rationale and operating procedures of indirect ELISA and capture ELISA.** (A) Indirect ELISA. (B) Capture ELISA. MSP, mosquito salivary protein. TMB, tetramethylbenzidine. oxTMB, oxidized tetramethylbenzidine. HRP, horseradish peroxidase. Fig 1 is created with BioRender.

highly abundant in mosquito saliva [18,19]. Therefore, we chose AAEL000749 as a representative MSP to demonstrate the effectiveness of the two ELISA methods. The titers of AAEL000749-specific IgE in mouse sera from one through four weeks post the repeated mosquito bites were measured. Serum samples from mice were diluted 1:10 for the assay. While the indirect ELISA failed to detect AAEL000749-specific IgE in all mouse sera (Fig 2B), capture ELISA was able to detect AAEL000749-specific IgE in the mouse sera collected at two through four weeks post mosquito exposure and showed that with increasing number and duration of mosquito exposure, the titers of AAEL000749-specific IgE in the mouse sera increased significantly (Fig 2C), indicating the capture ELISA had a significant higher detection sensitivity than the indirect ELISA.

Next, we investigated the specificity of the capture ELISA. The capture ELISA was used to measure the titers of AAEL000749-specific IgE in the sera of mice immunized with AAEL000749 protein (positive control), AAEL006347 protein

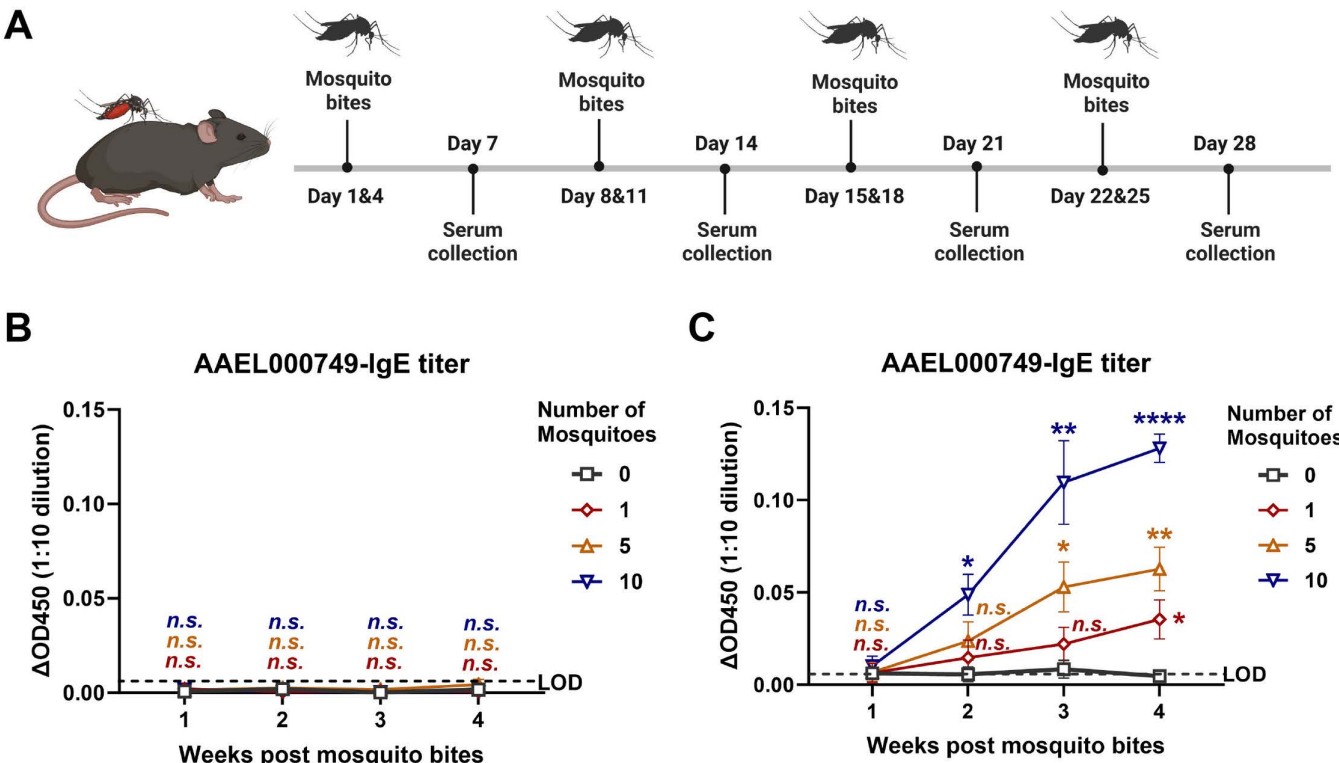

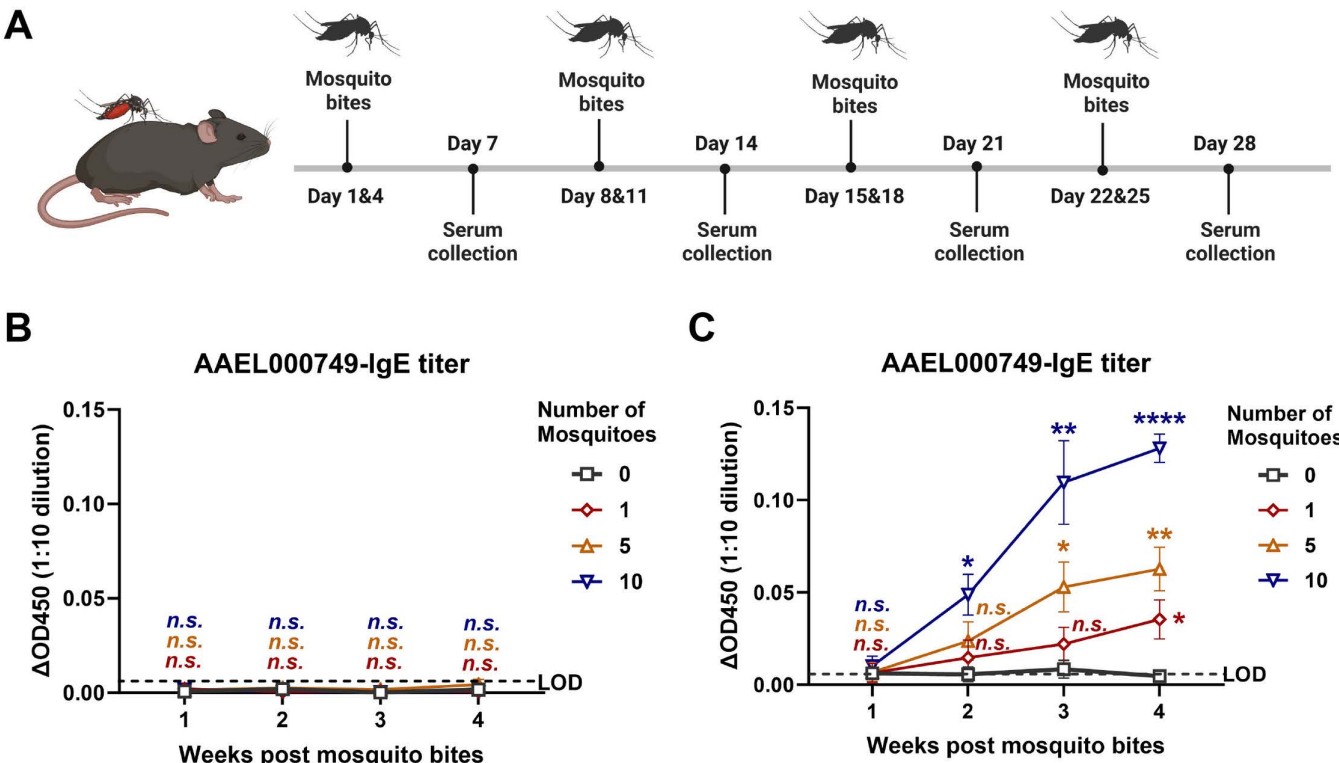

**Fig 2. Comparison of the effectiveness of indirect ELISA and capture ELISA in detecting mosquito salivary protein (MSP)-specific IgE in mouse serum.** (A) Schematic delineation of the mouse model of repeated mosquito exposure. C57BL/6J mice were exposed to bites from one, five, or ten uninfected *Aedes aegypti* mosquitoes per mouse, twice weekly, over a period of four consecutive weeks. The unbitten mice served as negative controls. Each group had six individual mice. Sera were collected at 7, 14, 21, and 28 days post the first mosquito bite. This schematic diagram was created with BioRender. (B-C) Detection of MSP AAEL000749-specific IgE in mouse sera by using the indirect ELISA (B) and the capture ELISA (C). The ΔOD450 for murine serum was calculated as the OD450 of each test sample (diluted 1:10) minus the mean OD450 of negative controls (also diluted 1:10). The limit of detection (LOD) was defined as twice the highest ΔOD450 among the negative controls. The number of mosquitoes indicates how many mosquitoes bit each mouse per exposure. Data are shown as mean ± SEM. Statistical analysis was performed using two-way ANOVA with multiple t-tests for post hoc comparisons. *n.s.*, not significant; *p < 0.05; **p < 0.01; ****p < 0.0001.

(an unrelated salivary protein control), or bovine serum albumin (BSA, a non-salivary protein control), and adjuvant-immunized mice. The purity and identity of MSPs AAEL000749 and AAEL006347 were confirmed by SDS-PAGE and WB (Fig 3A). AAEL000749-specific IgE was detected in the sera of all the mice immunized with AAEL000749 proteins but not in the sera of mice immunized with any other protein or adjuvant (Fig 3B), indicating that the capture ELISA had an excellent detection specificity.

## Comparison of the effectiveness of capture ELISA and indirect ELISA in detecting MSP-specific IgE in human serum

We recruited four healthy volunteers: Volunteers 1, 2, and 3 had no prior history of *Aedes aegypti* mosquito bites, while Volunteer 4 had occasional previous exposure to *Aedes aegypti* mosquitoes and therefore served as a positive control. We performed mosquito exposure on each volunteer twice a week for 4 weeks, with 10 uninfected *Aedes aegypti* mosquitoes biting the volunteers on the forearm for 5 min each time (Fig 4A). Serum samples were tested for AAEL000749-specific IgE before and one through four weeks post mosquito bite using both the indirect ELISA and the capture ELISA. Serum samples from Volunteers 1 and 4 tested positive for AAEL000749-specific IgE at two and four weeks post

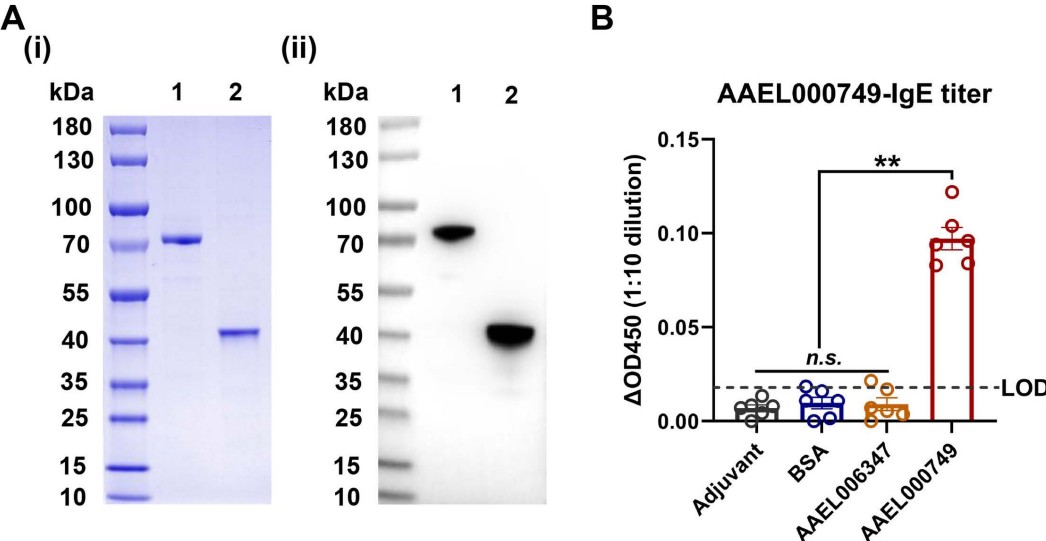

**Fig 3. The specificity of the capture ELISA.** (A) Validation of the purity and identity of recombinant MSPs AAEL0006347 and AAEL000749 by SDS-PAGE (i) and V5-tag targeted western blotting (ii), respectively. Lane 1, AAEL006347; Lane 2, AAEL000749. (B) Detection of AAEL000749-specific IgE in mouse sera. C57BL/6J mice were immunized twice at a two-week interval with *Aedes aegypti* salivary proteins AAEL000749, AAEL006347, or with BSA or adjuvant as controls. Each group consisted of six mice. Sera were collected at 7 days after the second immunization, and AAEL000749-specific IgE titers were measured using the capture ELISA. The ΔOD450 for murine serum was calculated as the OD450 of each test sample (1:10 dilution) minus the mean OD450 of negative controls (naïve, untreated mice; 1:10 dilution). The LOD was defined as twice the maximum ΔOD450 among the negative controls. Data are presented as mean ± SEM. Statistical analyses were performed using one-way ANOVA followed by multiple t-tests for post hoc comparisons. *n.s.*, not significant; **$p < 0.01$.

mosquito exposure, with an unexpected decrease in titer at four weeks compared to two weeks, as measured by indirect ELISA (Fig 4B). This suggests that the indirect ELISA lacks sensitivity and accuracy. However, the capture ELISA detected AAEL000749-specific IgE in the sera of all volunteers as early as two weeks post mosquito exposure, with titers increasing over time and with greater mosquito exposure (Fig 4C). As expected, Volunteer 4, who had prior *Aedes aegypti* exposure, showed significantly higher AAEL000749-specific IgE titers throughout the trial period compared to the other volunteers. These results again indicate that the capture ELISA has a much better detection sensitivity and accuracy than the indirect ELISA.

Next, we quantitatively assessed the detection limit, detection sensitivity, false positive rate, and false negative rate of our capture ELISA when used to detect *Aedes aegypti* salivary protein AAEL000749-specific IgE in human serum samples. First, we recruited 20 positive volunteers who had been bitten by *Aedes aegypti* mosquitoes and 20 negative volunteers who had not been bitten by the mosquitoes (S1 Table). The *Aedes aegypti* bite-positive volunteers consists of 20 severe dengue (SD) patients recruited in Xishuangbanna, China, in August 2024. In August, the population density of the mosquitoes in Xishuangbanna usually reaches its annual peak, and the intensity of dengue transmission is also correspondingly among the highest of the year [20,21]. In Xishuangbanna, dengue virus is primarily transmitted by *Aedes aegypti* bites [22,23]; therefore, the SD patients were theoretically repeatedly bitten by the mosquitoes (all volunteers self-reported a history of frequent mosquito bites before illness onset). Their acute-phase sera are theoretically expected to contain AAEL000749-specific IgE antibodies, and thus they can be considered an *Aedes aegypti* bite-positive population. We collected their acute-phase sera, and measured the titers of AAEL000749-specific IgE in their sera using our capture ELISA.

The results showed that, among the 20 volunteers who were positive for *Aedes aegypti* bites, all 20 had positive serum AAEL000749-specific IgE, and none were negative, indicating a test sensitivity of 100% (20/20) and a false negative rate of 0% (0/20) (S1 Table). Among the 20 negative volunteers, none had positive serum AAEL000749-specific IgE, and all

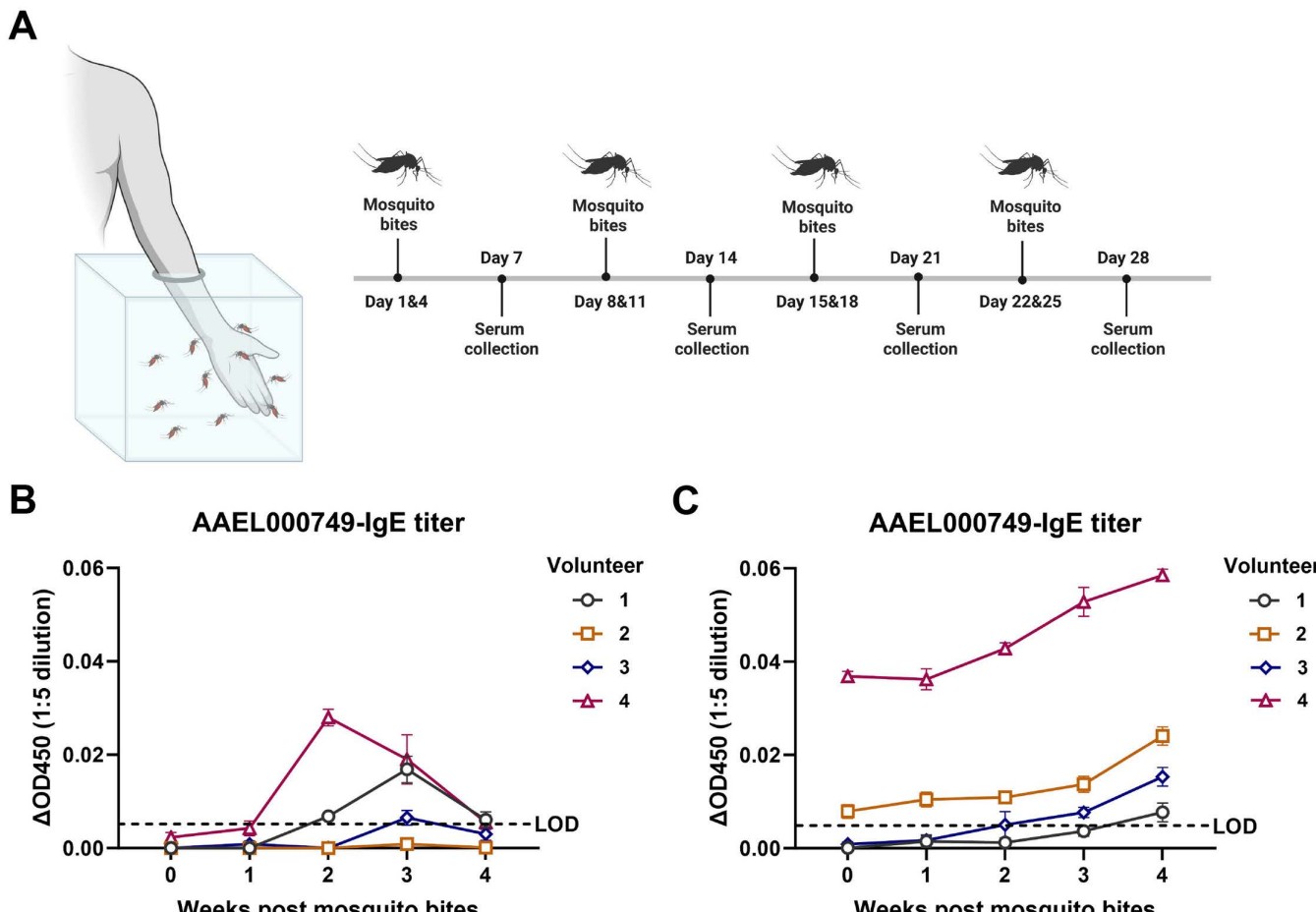

**Fig 4. Comparison of the effectiveness of capture ELISA and indirect ELISA in detecting MSP-specific IgE in human serum.** (A) Schematic delineation of the human trial of repeated mosquito exposure. Four healthy adult volunteers including Volunteer 1, 2, and 3 without- and Volunteer 4 with- prior history of *Aedes aegypti* mosquito bites were recruited. Each volunteer underwent controlled bites by ten uninfected *Aedes aegypti* mosquitoes twice weekly for four consecutive weeks. Sera were collected at 7, 14, 21, and 28 days post the first mosquito bite. This schematic diagram was created with BioRender. (B-C) Detection of MSP AAEL000749-specific IgE in human sera by using the indirect ELISA (B) and the capture ELISA (C). The ΔOD450 was calculated as the OD450 of each human serum sample (1:5 dilution) minus the mean OD450 of negative controls (eight volunteers never exposed to *Aedes aegypti*, 1:5 dilution). The LOD was defined as twice the maximum ΔOD450 among negative controls. Each serum sample was tested in quadruplicate, and results are expressed as mean±SEM.

20 were negative, resulting in a test specificity of 100% (20/20) and a false positive rate of 0% (0/20) (S1 Table). These results demonstrate that the capture ELISA we established possesses good sensitivity and specificity.

In addition, we pooled the serum from all volunteers who were positive for *Aedes aegypti* bites and designated this as the "positive serum" which represents the average level in individuals who are positive for *Aedes aegypti* bites in the general population. We serially diluted this serum and used the capture ELISA to detect the titer of AAEL000749-specific IgE at different dilutions. The detection limit of our assay is defined as the highest dilution at which AAEL000749-specific IgE is still detected as positive before it turns negative in the next dilution. In practice, the capture ELISA we developed can only provide a relative quantification of AAEL000749-specific IgE, and its detection limit is difficult to express in terms of the absolute content of AAEL000749-specific IgE due to the lack of a standard. However, the total IgE concentration in serum corresponding to this detection limit can be easily measured. Thus, we used the total IgE content at the detection

limit serum dilution (1:80, S2 Table) as a surrogate for the detection limit of AAEL000749-specific IgE, which was calculated to be 87.42 ng/mL, and the total IgE concentration in the positive serum was 6993.41 ng/mL (S2 Table).

**Application of the capture ELISA to monitor the AAEL000749-specific IgE in the sera of the healthy people living in the areas where *Aedes aegypti* widely distributed**

Finally, we used the capture ELISA to dynamically monitor the titers of AAEL000749-specific IgE in the sera of healthy people in Xishuangbanna where *Aedes aegypti* is widely distributed [21,23]. We graded the titers of AAEL000749-specific IgE in sera as high, mediate, low, and negative according to the corresponding ΔOD450 value (Fig 5A). As expected, the proportions of high-, mediate-, and low-titer AAEL000749-specific IgE populations in August, September, and October were significantly higher than in January, February, and March (Fig 5B), probably due to the greater mosquito population

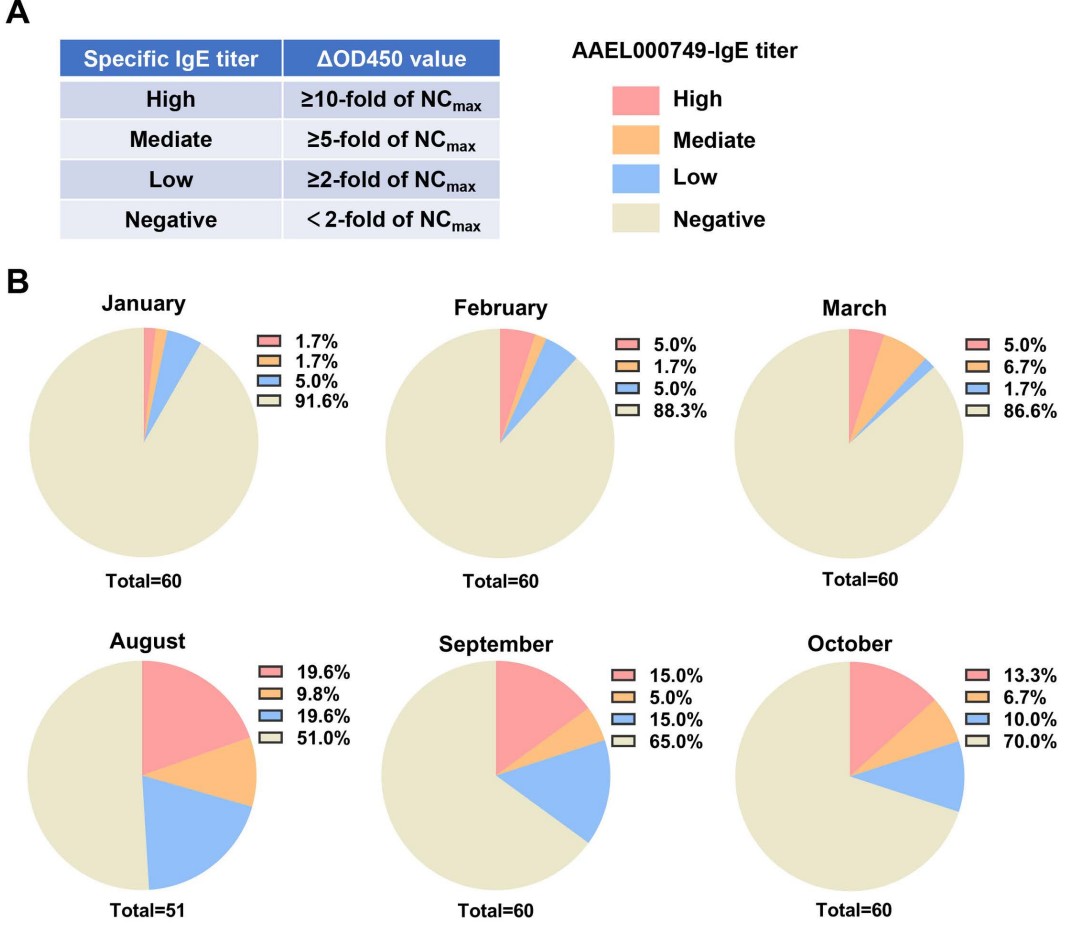

**Fig 5. Detection of AAEL000749-specific IgE in the sera of healthy individuals residing in areas with widespread *Aedes aegypti* distribution.** (A) Grading criteria for AAEL000749-specific IgE. $NC_{max}$, the maximum ΔOD450 among the negative controls. AAEL000749-specific IgE was detected by the capture ELISA. The ΔOD450 value was calculated as the OD450 of each human serum sample (diluted 1:5) minus the mean OD450 of negative controls (serum from eight *Aedes aegypti*-unexposed volunteers, also diluted 1:5). (B) Proportions of healthy individuals in Xishuangbanna exhibiting varying grades of AAEL000749-specific IgE titers across different months. In the months with lower mosquito population density (January, February, and March), 60 volunteers were included each month. For the months with higher mosquito density (August, September, and October), the numbers of volunteers were 51, 60, and 60, respectively. The proportions of healthy volunteers with high (red), medium (yellow), and low (blue) titers of AAEL000749-specific IgE, as well as seronegative individuals (gray), are shown to the right of the corresponding figure legend.

density during the former months [21,24]. This result again suggested that our capture ELISA can be a powerful tool to monitor the MSP-specific IgE in human serum.

## Discussion

Mosquito salivary proteins (MSPs) possess various physiological functions, including facilitating the ingestion of plant sap and blood, defending against microbial infections, and initiating the digestion of food, all of which are essential for mosquito survival [25]. In addition, MSPs are key mediators of mosquito-host interactions, significantly influencing both local and systemic immune responses in the host, as well as modulating viral infection at the bite site and subsequent systemic dissemination [25]. These proteins are highly immunogenic and can induce specific humoral and cellular immune responses in the host [6,26]. Such MSP-specific immune responses not only have a profound impact on the transmission of mosquito-borne viruses [27–30], but also trigger varying degrees of mosquito allergic reactions, and in severe cases, this may result in mosquito allergy syndrome, greatly affecting patient health and quality of life [4]. MSPs induced IgE plays a pivotal role in mediating such allergic reactions [31]. Thereafter, detection of MSP-specific IgE in host serum is of great significance.

Mosquito bites induce the host producing MSP-specific IgG, IgE, and other isotypes of immunoglobulin, among which IgG exhibits significantly higher concentration than IgE [32], thereby strongly interfering with the detection of MSP-specific IgE in serum. At present, the literature describes two primary approaches for detecting MSP-specific IgE: ELISA-based assays [10–12] and immunoblotting-based assays [6–8]. Most ELISA approaches embed MSPs on the microplate bottom, followed by direct incubation with serum samples [10–12]. This format suffers from a major limitation that the much more abundant IgG in serum competitively interferes with IgE binding, substantially reducing assay sensitivity [33]. Immunoblotting involves separating MSPs by gel electrophoresis, transferring them onto membranes, then incubating diluted serum with the membrane and detecting MSP-specific IgE using anti-IgE antibodies [6–8]. However, this method also faces IgG interference and does not allow accurate quantification of IgE titers in serum. Brummer-Korvenkontio et al. [15] reported an IgE-capture ELISA for detecting MSP-specific IgE, where IgE-capture antibodies were coated on the microplate to selectively enrich IgE from serum samples. Other immunoglobulins such as IgG and IgM were washed away during the process. This strategy not only improves specificity but also significantly enhances sensitivity by eliminating interference from non-IgE immunoglobulins. However, the method still has space for improvement. The capture ELISA also necessitates gathering a substantial quantity of MSPs [15], which is a laborious and time-consuming work. In addition, the anti-MSPs serum prepared from mosquito bitten rabbit may not be as specific or efficient as a V5-tag monoclonal antibody for the detection of MSPs.

To address the above shortcomings, we developed a more powerful capture ELISA. Firstly, we coated the microplates with a monoclonal antibody against the Fc region of IgE, which ensures more specific enrichment of serum IgE. Because IgE is captured via the Fc region, the antigen-binding capacity of the F(ab)$_2$ region remains unaffected. Secondly, we replaced the labor-intensive collection of natural MSPs with highly purified, *Drosophila* S2 cell-expressed recombinant MSPs. This enables the specific detection of IgE against an individual MSP, rather than pan-IgE against all MSPs. Furthermore, a V5 tag was fused to the C-terminus of each recombinant MSP, allowing the use of a single V5 tag-specific IgG antibody for detection of all recombinant MSPs, thus eliminating the requirement to develop a specific antibody for each individual MSP. We used a commercially available V5 monoclonal antibody instead of the rabbit anti-MSP serum used by Brummer-Korvenkontio, thereby achieving higher detection specificity. Finally, we used an HRP-conjugated secondary antibody for signal amplification and an ultrasensitive TMB substrate for colorimetry, both of which could enhance the assay sensitivity.

In this study, we selected IgE specific to AAEL000749, a highly abundant mosquito salivary protein [18,19], as the target for detection to compare the performance of our novel capture ELISA with the conventional indirect ELISA widely used in the literature. The results demonstrated that the novel capture ELISA efficiently and specifically detected

AAEL000749-specific IgE in both mouse sera and human sera from repeated mosquito exposure. In contrast, the indirect ELISA failed to do so. Thus, the novel capture ELISA showed superior sensitivity and specificity. Furthermore, we applied the capture ELISA to measure AAEL000749-specific IgE titers in the sera of healthy individuals from Xishuangbanna, a region with widespread *Aedes aegypti* distribution [34]. As expected, the IgE titers were significantly higher in months with greater mosquito population density compared to months with lower density. These findings indicate that the novel capture ELISA enables efficient and specific detection of serum IgE against individual MSPs, and can serve as a powerful tool for monitoring MSP-specific IgE levels in human populations.

However, the capture ELISA also has several limitations, which are summarized as follows: 1) In natural environments, humans may be bitten by more than one mosquito species. Certain salivary proteins from different mosquito species are evolutionarily conserved and may possess similar or identical antigenic epitopes. This may result in potential cross-reactivity between these proteins and their specific IgE antibodies, thereby affecting the specificity of the assay. In practical applications, if information on mosquito species is relevant, it is advisable to simultaneously test for IgE specific to homologous salivary proteins from other locally prevalent mosquito species, in addition to the target species. By comparing the results, the dominant mosquito species responsible for the specific IgE response can be identified. 2) IgE is present at a very low concentration in host serum, and the quantity of IgE specific to a certain MSP is even lower. Thus, the OD450 values obtained from either conventional indirect ELISA or capture ELISA are generally quite low. This is in sharp contrast to the detection of IgG, which is abundant in serum and yields much higher OD450 values. Therefore, attention must be paid to this issue. One optimization strategy is to dilute the tested sera as little as possible, for example at 1:2 or 1:5, to maximize the amount of total IgE captured and thereby increase the OD450 value for MSP-specific IgE. 3) Compared with conventional indirect ELISA, the capture ELISA involves more experimental steps, requires additional reagents, and is more time-consuming, which are its inherent drawbacks. To shorten the experimental time, steps such as coating with capture antibody, blocking, sample incubation, and incubation with antigen and antibodies can be performed at 37°C, which can considerably accelerate the overall procedures.

In summary, the novel capture ELISA is a powerful tool for detection of MSP-specific IgE in both mouse and human serum. In view of the excellent detection performance, this assay merits promotion for future applications, such as the identification of specific allergens in mosquito saliva, the diagnosis and prognosis of mosquito allergy, and epidemiological surveillance of mosquito exposure levels in different populations.

## Supporting information

**S1 Table. AAEL000749-specific IgE titers in the sera of mosquito bite-positive and -negative populations using the capture ELISA.**
(DOCX)

**S2 Table. AAEL000749-specific IgE titers and total IgE concentration in *Aedes aegypti* bite-positive pooled human serum.**
(DOCX)

## Acknowledgments

We thank Dr. Liu Yang and Nurse Xianmei Zhou from Xishuangbanna Prefecture People's Hospital for their contributions to the collection of volunteers' serum samples. We also express our gratitude to all the volunteers who participated in this study for their dedication to the advancement of medical science.

## Author contributions

**Conceptualization:** Zhaoyang Wang, Gong Cheng.

**Data curation:** Zhaoyang Wang, Fen Zeng.

**Formal analysis:** Zhaoyang Wang.

**Investigation:** Zhaoyang Wang, Yan Liang.

**Methodology:** Zhaoyang Wang.

**Project administration:** Tingting Li, Gong Cheng.

**Resources:** Fen Zeng, Tingting Li.

**Supervision:** Tingting Li, Gong Cheng.

**Validation:** Zhaoyang Wang, Yan Liang.

**Visualization:** Zhaoyang Wang, Yan Liang.

**Writing – original draft:** Zhaoyang Wang.

**Writing – review & editing:** Zhaoyang Wang, Gong Cheng.

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
