## [Decision Letter · Decision Letter 0]

1 Jul 2025

PNTD-D-25-00667

A capture enzyme-linked immunosorbent assay for detection of mosquito salivary protein-specific immunoglobulin E

Dear Dr. Cheng,

Thank you for submitting your manuscript to PLOS Neglected Tropical Diseases. After careful consideration, we feel that it has merit but does not fully meet PLOS Neglected Tropical Diseases's publication criteria as it currently stands. Therefore, we invite you to submit a revised version of the manuscript that addresses the points raised during the review process.

Please submit your revised manuscript within 60 days Aug 30 2025 11:59PM. If you will need more time than this to complete your revisions, please reply to this message or contact the journal office at plosntds@plos.org. Please include the following items when submitting your revised manuscript:

We look forward to receiving your revised manuscript.

Kind regards,

Adly M.M. Abd-Alla, Prof asso.

Section Editor

Adly Abd-Alla

Section Editor

Shaden Kamhawi

co-Editor-in-Chief

Paul Brindley

co-Editor-in-Chief

**Journal Requirements:**

At this stage, the following Authors/Authors require contributions: Zhaoyang Wang, Yan Liang, Fen Zeng, Tingting Li, and Gong Cheng. Please ensure that the full contributions of each author are acknowledged in the "Add/Edit/Remove Authors" section of our submission form.

2) Some material included in your submission may be copyrighted. According to PLOSu2019s copyright policy, authors who use figures or other material (e.g., graphics, clipart, maps) from another author or copyright holder must demonstrate or obtain permission to publish this material under the Creative Commons Attribution 4.0 International (CC BY 4.0) License used by PLOS journals. Please closely review the details of PLOSu2019s copyright requirements here: PLOS Licenses and Copyright. If you need to request permissions from a copyright holder, you may use PLOS's Copyright Content Permission form.

Potential Copyright Issues:

- Figures 2 and 4.. Please confirm whether you drew the images / clip-art within the figure panels by hand. If you did not draw the images, please provide (a) a link to the source of the images or icons and their license / terms of use; or (b) written permission from the copyright holder to publish the images or icons under our CC BY 4.0 license. Alternatively, you may replace the images with open source alternatives. See these open source resources you may use to replace images / clip-art:

3) Please ensure that the funders and grant numbers match between the Financial Disclosure field and the Funding Information tab in your submission form. Note that the funders must be provided in the same order in both places as well.

**Reviewers' Comments:**

Reviewer's Responses to Questions

**Key Review Criteria Required for Acceptance?**

**Methods**

-Are the objectives of the study clearly articulated with a clear testable hypothesis stated?

-Is the study design appropriate to address the stated objectives?

-Is the population clearly described and appropriate for the hypothesis being tested?

-Is the sample size sufficient to ensure adequate power to address the hypothesis being tested?

-Were correct statistical analysis used to support conclusions?

-Are there concerns about ethical or regulatory requirements being met?

Reviewer #1: - The objectives of the study are clearly stated

- The study design is appropriate concerning the objective of method development, but comparisons to other studies should be toned down, and certain parameters in study design and results must be reported with much more technical detail to be able to judge on that.

- Study population is well described.

- Sample sizes are sufficient.

- Statistical analysis was performed.

- The authors list the permissions of Ethical Committee.

Reviewer #2: This study presents clearly formulated, testable hypotheses and employs a rigorous experimental design to validate the research aims. The participant population is precisely defined and appropriate for the stated hypotheses, with a sample size ensuring statistical adequacy. Data were analyzed using suitable statistical approaches, and the conclusions are well-supported by empirical evidence. The research complies fully with ethical review and regulatory requirements.

Reviewer #3: See below

**Results**

-Does the analysis presented match the analysis plan?

-Are the results clearly and completely presented?

-Are the figures (Tables, Images) of sufficient quality for clarity?

Reviewer #1: - The results are currently not presented well enough, especially 3B and C and 4B and C: ELISA units miss explanation, the signal looks very low at the moment. More technical details should be given on antigen characterization to support the claims in the comments.

- The figures are of sufficient resolution (but should be more informative).

Reviewer #2: The results are clearly presented, and the image quality meets publication standards.

Reviewer #3: See below

**Conclusions**

-Are the conclusions supported by the data presented?

-Are the limitations of analysis clearly described?

-Do the authors discuss how these data can be helpful to advance our understanding of the topic under study?

-Is public health relevance addressed?

Reviewer #1: At the moment, the conclusions are not supported well because certain methodology steps and data presentation require more information as available now. The authors also compare their work with other published methods, but the comparisons are not based on any parameter, and do not show direct experimental evidence by which the judgement could be made. The limitations are not discussed. Public health relevance is addressed.

Reviewer #2: The conclusions of this study are derived from the analysis of experimental data, with the limitations explicitly discussed in the discussion section. The findings demonstrate potential applications in disease screening.

Reviewer #3: See below

**Editorial and Data Presentation Modifications?**

Reviewer #1: (No Response)

Reviewer #2: Minor Revision

Reviewer #3: See below

**Summary and General Comments**

Reviewer #1: In the present manuscript, the authors describe the construction of a capture ELISA assay for detection of IgE directed against mosquito salivary protein AAEL000749. For this purpose, they first produce the salivary protein in insect cells, and purify it via cobalt affinity chromatography. Then they use it for mouse immunization, to obtain the sera to demonstrate the specificity of the ELISA. Their proposed ELISA uses the therapeutic antibody Omalizumab for specific IgE capture, recombinant V5-labelled antigen as a bait, and finally anti-V5 tag antibody and secondary HRP-labelled antibody for detection. Indeed, anti-AAEL000749 -immunized mice sera are the only ones testing positive, when examined along with sera from adjuvant - only, BSA, or AAEL6347 (unrelated mosquito saliva protein)-immunized mice. They the perform animal experiment and expose mice to mosquito bites, and measure IgE in mouse serum 4 weeks post mosquito bites. With a greater number of mosquitos, and continuing exposure, also the IgE titer in serum increases. This specific IgE can only be detected by Capture ELISA, but less with Indirect ELISA (at least as presented here). Also, healthy human volunteers were exposed to mosquito bites in a similar way, and Capture ELISA shows more sensitive capture of the AAEL000749-specific IgE than the indirect assay. Finally, the authors measure the antigen-specific IgE in 60 sera, and estimate the reactivity in fold of negative control, and the sera appear to have a higher content of AAEL000749-IgE in the months August-October than in January-March.

Methods that facilitate the estimation of the mosquito bite-related IgE are important because this value can inform on potential allergic response if the bites repeat. However, there is a number of questions and issues that the authors should address:

- The authors claim that the ELISA they present is of a higher sensitivity and specificity than other reported methods, but the side-to-side comparison with other methods is not really shown, and sensitivity or specificity (i.e. detection limit / number of false positives) are not shown or estimated.

- The authors claim that the recombinant antigens produced here should be superior to bacterial, but there is not really a direct comparison shown, using either any biophysical method, or control antibody reactivity, or biological activity.

- Labels of the Figures and Legend text are currently not sufficient for understanding (please see specific remarks below), and the value of their developed test is indeed difficult to judge until they provide more information.

- The abstract describes the methodology – I propose the findings of the study are highlighted instead.

- Several passages in the text repeat twice or three times, and I suggest that these redundancies are removed.

Please find below a list of remarks which I hope will be helpful.

Lines 27-37: “Current enzyme-linked immunosorbent assay (ELISA) and immunoblotting methods for detecting MSP specific IgE suffer from interference by abundant, high-affinity IgG, leading to low sensitivity.“ – there is no direct comparison with current methods shown in this manuscript

Line 89:“ with nice detection sensitivity“- this is colloquial language, please reword to “high detection sensitivity”, or similar

Line 93: “specific IgG will preferentially bind to the embedded MSPs due to its superior content and affinity“. This sentence would be better with another word-order: immobilized MSPs will preferentially be occupied by the specific IgG due to its superior concentration. “superior affinity” – can such statement be generalized and could you provide a reference?

Line 97:” needs to collect“ – requires the use of

Line 100: “used in the current literature“ – described in the current literature

Line 104: “MSPs expressed in Drosophila S2 cells…” – this is a very generalized statement, and no evidence for this claim is presented. Additionally, reference 14 only shows the results for recombinantly expressed Aed al 2, Aed al 13 and Aed al 14, and these should not be generalized as MSPs.

Line 111: “The highly pure MSPs…“ this manuscript does not show this data.

Line 181: purification procedure (buffers) and protein measurement and storage should be described briefly.

Line 192: “goat anti-mouse IgE secondary antibody“ – please use the RRID identificators for commercially acquired antibodies (and please throughout the manuscript).

Line 268: Figure 2 B, C - the OD 450 is low here. What does OD 450 (1:10 dilution) on the y-axis mean, is this serum dilution? Figure legend (0, 1, 5 or 10 mosquitos) should be explained in the text form.

Line 290: “Purity and identity” – western blot is performed with an anti-V5 antibody, so it does not give information on the identity of the protein.

Figure 4: Is the value on Y axis the actual OD 450 – the values are low for ELISA. What does OD 450 (1:5 dilution) on the y-axis mean, is this serum dilution? How can the U-shape of the curve in 4B be explained – is it possible that the results would look different with another serum dilution?

Line 363: This is also a very interesting example of the AAEL000749 activity: https://pmc.ncbi.nlm.nih.gov/articles/PMC5025043/.

Line 350: NC-is negative control, but what was the negative control used here.

Line 370: Reference 25 uses different antigens, not all antigens used were similar in antibody induction for human and mice. Which data indicate that IgG produced as a response is of higher affinity?

Line 371: “Currently, two primary methods are reported…“ – this sentence repeats from the introduction

Line 386:” However, the method still has space for improvement“ – the novel method described here and the referenced one should be compared side-to-side to confirm this.

Line 387:” collect a large amount of MSPs“ - this sentence repeats from the introduction

Line 388:” serum prepared from mosquito bitten rabbit may not be so specific or efficient “ – why would other mosquito-bitten species produce serum that is more specific or efficient?

Line 405: “selected IgE specific to AAEL000749“ – the specificity should be demonstrated.

Line 419: for detection of MSP-specific

Reviewer #2: In this manuscript, Wang et al. present an innovative and highly efficient capture ELISA designed to address these limitations. The assay utilizes microplates coated with a monoclonal antibody targeting the IgE Fc region, thereby enriching specifically for IgE without affecting the antigen-binding F(ab)2 region. Instead of natural MSPs, recombinant MSPs expressed in Drosophila S2 cells were employed, enabling the detection of IgE responses to individual MSPs. Each recombinant MSP includes a C-terminal V5 tag, facilitating detection with a V5-specific monoclonal antibody, which improves specificity and streamlines the overall assay workflow. Signal amplification was achieved through the use of an HRP-conjugated secondary antibody and an ultrasensitive TMB substrate. Overall, this capture ELISA offers a highly specific and sensitive method for the detection of MSP-specific IgE in human and mouse serum samples. Moreover, detecting other antibody types, such as IgM, IgE, or IgA, which typically have significantly lower affinity and concentrations compared to IgG, faces a common challenge: interference from abundant IgG that cannot be ignored. The capture ELISA method proposed by Wang et al. may also serve as a valuable reference for the detection of antigen-specific IgE, IgM, or IgA.

Reviewer #3: Dear Editor and authors,

Thank you for the opportunity to review the manuscript titled "A capture enzyme-linked immunosorbent assay for detection of mosquito salivary protein-specific immunoglobulin E" (PNTD-D-25-00667). This research article presents a significant advancement in the detection of mosquito salivary protein (MSP)-specific IgE, addressing critical limitations of current methods. Overall, this is a very strong manuscript. The authors have developed a novel capture ELISA method that demonstrates superior sensitivity and specificity compared to traditional indirect ELISA. The work is well-designed, executed, and clearly presented. The findings have important implications for understanding mosquito allergy and for epidemiological surveillance. The writing quality of the manuscript is excellent, demonstrating a high standard of formal scientific English. However, I have identified a few areas where minor improvements could further enhance its clarity and completeness:

• In the abstract, the phrase "allowing detection with a V5-specific monoclonal antibody, enhancing specificity and" is repeated. Consider rephrasing for better conciseness. For example, the sentence could be streamlined to: "Each recombinant MSP carried a C-terminal V5 tag, enabling detection with a V5-specific monoclonal antibody, thereby enhancing specificity and simplifying the assay."

• The discussion effectively highlights the advantages of the developed capture ELISA over existing methods. However, a dedicated, explicit paragraph or sentence discussing any inherent limitations of this specific capture ELISA would strengthen the manuscript. For instance, consider addressing aspects such as: Potential for cross-reactivity with other IgE types, if any, despite the Fc region capture; Cost-effectiveness or throughput considerations for very large-scale screening, if applicable; and Any specific challenges encountered during its development or application that were successfully overcome but are worth noting for future researchers.

• The concluding sentence in the discussion, "In summary, the novel capture ELISA is a powerful tool for detection MSP-specific IgE in both mouse and human serum and deserves well promoting," is concise. However, briefly expanding on what "deserves well promoting" entails could be beneficial. What are the next logical steps for this assay? For example, its potential application in: Clinical diagnostics for mosquito allergy; Broader epidemiological studies to map IgE sensitization in different populations; Assessment of vaccine efficacy against mosquito-borne diseases; and Further characterization of specific MSPs as biomarkers.

• Consider using the STARD protocol to describe your findings.

• Describe the coating buffer used in the immunoassays.

PLOS authors have the option to publish the peer review history of their article (what does this mean?). If published, this will include your full peer review and any attached files.

Reviewer #1: No

Reviewer #2: No

Reviewer #3: **Yes: **FRED LUCIANO NEVES SANTOS

**Figure resubmission:**
---

## [Decision Letter · Decision Letter 1]

12 Aug 2025

Dear Dr. Cheng,

We are pleased to inform you that your manuscript 'A capture enzyme-linked immunosorbent assay for detection of mosquito salivary protein-specific immunoglobulin E' has been provisionally accepted for publication in PLOS Neglected Tropical Diseases.

Best regards,

Adly M.M. Abd-Alla, Prof asso.

Section Editor

Adly Abd-Alla

Section Editor

Shaden Kamhawi

co-Editor-in-Chief

Paul Brindley

co-Editor-in-Chief

Reviewer's Responses to Questions

**Key Review Criteria Required for Acceptance?**

**Methods**

-Are the objectives of the study clearly articulated with a clear testable hypothesis stated?

-Is the study design appropriate to address the stated objectives?

-Is the population clearly described and appropriate for the hypothesis being tested?

-Is the sample size sufficient to ensure adequate power to address the hypothesis being tested?

-Were correct statistical analysis used to support conclusions?

-Are there concerns about ethical or regulatory requirements being met?

Reviewer #1: In the revised version of the manuscript, the objectives are stated more clearly and experimental set-ups are described in more detail. Outcomes of statistical analysis are better presented. Ethics statement confirming the study approval is presented.

Reviewer #2: The authors have provided a comprehensive and satisfactory response to the reviewers' comments. The newly added analyses directly and effectively addressed the previously raised methodological concerns, significantly enhancing the credibility of the study findings and the robustness of the conclusions. These analyses are not superficial but provide substantive value, resulting in a methodologically stronger paper.

Reviewer #3: See below

**Results**

-Does the analysis presented match the analysis plan?

-Are the results clearly and completely presented?

-Are the figures (Tables, Images) of sufficient quality for clarity?

Reviewer #1: The results are clearly presented,, Figures and tables are of sufficient quality.

Reviewer #2: The analyses conducted by the authors are consistent with the analysis plan. The results are clearly and completely presented, and the quality of the figures (tables, images) is sufficiently clear.

Reviewer #3: See below

**Conclusions**

-Are the conclusions supported by the data presented?

-Are the limitations of analysis clearly described?

-Do the authors discuss how these data can be helpful to advance our understanding of the topic under study?

-Is public health relevance addressed?

Reviewer #1: The conclusions are well presented, the advancements that the study brings are realistically pointed out, and the limitations are mentioned. The study is very relevant for public health, for diagnosis as for pandemic preparedness, which is also well described.

Reviewer #2: The authors have supplemented new data that further supports the conclusions. Additionally, the discussion section describes the limitations of the analyses. This data helps advance our understanding of public health.

Reviewer #3: See below

**Editorial and Data Presentation Modifications?**

Reviewer #1: (No Response)

Reviewer #2: accept

Reviewer #3: See below

**Summary and General Comments**

Reviewer #1: The authors have addressed all the concerns from the first round of revision and the manuscript was substantially improved.

Reviewer #2: This study presents a well-executed and timely investigation into specific IgE ELISA testing. The authors have successfully addressed all key methodological concerns raised during peer review through rigorous supplementary analyses and detailed clarifications, markedly strengthening the validity and robustness of their findings.

Reviewer #3: Dear Editor,

I have carefully evaluated the revised version of the manuscript. The authors have addressed all my comments and have incorporated the suggested changes into the final text. The revisions have improved the clarity, accuracy, and overall quality of the manuscript.

In my opinion, the manuscript is now suitable for publication in its current form.

PLOS authors have the option to publish the peer review history of their article (what does this mean?). If published, this will include your full peer review and any attached files.

Reviewer #1: No

Reviewer #2: No

Reviewer #3: **Yes: **FRED LUCIANO NEVES SANTOS

---

## [Editor Report · Acceptance letter]

Dear Dr. Cheng,

We are delighted to inform you that your manuscript, " 

A capture enzyme-linked immunosorbent assay for detection of mosquito salivary protein-specific immunoglobulin E," has been formally accepted for publication in PLOS Neglected Tropical Diseases.

Best regards,

Shaden Kamhawi

co-Editor-in-Chief

Paul Brindley

co-Editor-in-Chief
